# Effects of Acute 2,3,7,8-Tetrachlorodibenzo-p-Dioxin Exposure on the Circulating and Cecal Metabolome Profile

**DOI:** 10.3390/ijms222111801

**Published:** 2021-10-30

**Authors:** Nicholas Dopkins, Wurood Hantoosh Neameh, Alina Hall, Yunjia Lai, Alex Rutkovsky, Alexa Orr Gandy, Kun Lu, Prakash S. Nagarkatti, Mitzi Nagarkatti

**Affiliations:** 1Department of Pathology, Microbiology and Immunology, University of South Carolina School of Medicine, Columbia, SC 29209, USA; nid4009@med.cornell.edu (N.D.); wurood814@gmail.com (W.H.N.); alina.hall@uscmed.sc.edu (A.H.); acrutkovsky@gmail.com (A.R.); alexa.gandy@uscmed.sc.edu (A.O.G.); prakash@mailbox.sc.edu (P.S.N.); 2Department of Environmental Sciences and Engineering, University of North Carolina, Chapel Hill, NC 27599, USA; lai7@live.unc.edu (Y.L.); kunlu@unc.edu (K.L.)

**Keywords:** 2,3,7,8-tetrachlorodibenzo-p-dioxin, toxicity, NFκB, aryl hydrocarbon receptor, metabolome, immunity

## Abstract

2,3,7,8-tetrachlorodibenzo-p-dioxin (TCDD) is a polyhalogenated planar hydrocarbon belonging to a group of highly toxic and persistent environmental contaminants known as “dioxins”. TCDD is an animal teratogen and carcinogen that is well characterized for causing immunosuppression through activation of aryl hydrocarbon receptor (AHR). In this study, we investigated the effect of exposure of mice to an acute dose of TCDD on the metabolic profile within the serum and cecal contents to better define the effects of TCDD on host physiology. Our findings demonstrated that within the circulating metabolome following acute TCDD exposure, there was significant dysregulation in the metabolism of bioactive lipids, amino acids, and carbohydrates when compared with the vehicle (VEH)-treated mice. These widespread changes in metabolite abundance were identified to regulate host immunity via modulating nuclear factor-kappa B (NF-κB) and extracellular signal-regulated protein kinase (ERK1/2) activity and work as biomarkers for a variety of organ injuries and dysfunctions that follow TCDD exposure. Within the cecal content of mice exposed to TCDD, we were able to detect changes in inflammatory markers that regulate NF-κB, markers of injury-related inflammation, and changes in lysine degradation, nicotinamide metabolism, and butanoate metabolism, which collectively suggested an immediate suppression of broad-scale metabolic processes in the gastrointestinal tract. Collectively, these results demonstrate that acute TCDD exposure results in immediate irregularities in the circulating and intestinal metabolome, which likely contribute to TCDD toxicity and can be used as biomarkers for the early detection of individual exposure.

## 1. Introduction

TCDD is a toxic and environmentally persistent contaminant that is produced as a by-product of industrial combustion processes, which garnered historical notoriety due to being a major component of the controversial herbicide used during the Vietnam War known as “Agent Orange” [1,2]. The pathology of TCDD exposure has been characterized by symptoms of immunotoxicity, hormonal dysregulation, developmental problems, increased cancer incidence, and local skin irritation in the form of chloracne [3,4,5,6,7,8]. TCDD toxicity is due to activation of the cytosolic receptor and transcription factor known as the AHR, which then enhances transcription of genes proximal to dioxin response elements (DREs) in the host genome [9,10]. Additionally, TCDD exposure leads to dysregulation of epigenetic regulators, such as the miRNA expression profile and microbiome composition, influencing downstream changes in phenotype [11,12,13,14]. Previous studies have demonstrated that the effects of TCDD are entirely dependent on the activation of the AHR, as AHR knockout mice are insensitive to the toxic effects of TCDD [15,16]. Experiments studying the effects of AHR ligand binding have demonstrated the promiscuous role the receptor plays in regulating cell cycle progression, cellular metabolism, and the terminal differentiation of immune cell subsets [17]. TCDD exploits AHR as a high-affinity ligand that, when bound, results in dysregulation of metabolism, immunity, and cell cycle progression [18,19,20,21]. 

Unlike TCDD, other dietary and endogenously produced AHR ligands also activate AHR but are involved in normal immune system homeostasis and physiological functions. The precise mechanisms of such discrepancy in action mediated by AHR ligands are unclear but may be associated with the degree of AHR affinity and the relative half-life of the ligand within the exposed host [22]. A large portion of the uncertain changes in host physiology following TCDD exposure can be attributed to a limited number of studies dedicated to studying the direct changes within the metabolomic profile immediately following TCDD exposure [23]. In the current study, therefore, we investigated the significant changes in metabolic activity within the GI tract and serum following acute TCDD exposure in naive mice. The timely importance of this study is derived from previous studies demonstrating that the effects of planar hydrocarbons on host physiology are often dependent on changes in the metabolomic profile of host organisms following AHR activation [24,25,26,27,28]. The goals of the current study were to identify key metabolites dysregulated immediately following acute TCDD exposure in order to provide a comprehensive background for the mechanisms by which TCDD instigates pathogenic effects related to carcinogenesis, metabolic disorders, and immunotoxicity. To the best of the authors’ knowledge, this is the first study focusing on the immediate changes in the metabolome following acute TCDD exposure.

## 2. Results

### 2.1. Characterization of Alterations in the Metabolic Profile Following Acute TCDD Exposure

To study the metabolic profile after TCDD exposure, mice were injected i.p. with TCDD or vehicle, and 72 h later, serum and cecal samples were analyzed. Within the serum of TCDD-exposed mice, there were 1212 confirmed metabolites identified, and of these metabolites, 299 reached the previously defined threshold of significance (Figure 1a). PCA scores plot comparing normalized peak intensities revealed distinct clusters separating the serum metabolomes of the VEH- and TCDD-exposed groups (Figure 1b). Within the cecal content, 902 confirmed metabolites were identified, and of these, 107 were significantly different between groups (Figure 1c). PCA scores plot for cecal metabolites revealed distinct clusters separating the two groups with some overlap between the VEH- and TCDD-exposed groups (Figure 1d). 

### 2.2. Identification of Metabolic Pathways Most Impacted Following Acute TCDD Exposure

MetaboAnalyst 3.0 revealed that within the serum, there were 404 Kyoto Encyclopedia of Genes and Genomes (KEGG) metabolic pathways that contained one or more significant metabolites. Pathway significance and impact were plotted on the pathway overview with larger, more intense red color circles indicating those overarching pathways were most meaningfully altered between groups (Figure 2a). Of these pathways, overlapping and incomplete pathways were identified, consolidated, and filtered, respectively, yielding 15 pathways with significant differences between TCDD and controls groups. In no significant order, these pathways within the serum involved the biosynthesis of the unsaturated fatty acids (FAs) (Appendix A), phenylalanine, tyrosine, and tryptophan biosynthesis (Appendix A), purine metabolism (Appendix A), lysine degradation (Appendix A), valine, leucine, and isoleucine metabolism (Appendix A), arginine biosynthesis (Appendix A), and sphingolipid metabolism (Appendix A). Cecal content pathway analysis yielded nine pathways containing one or more significantly modulated metabolites, although both significance and impact of these pathways were reduced relative to serum pathway analysis (Figure 2b). In no significant order, these pathways within the cecal contents involved purine metabolism (Appendix A), lysine degradation (Appendix A), nicotinate and nicotinamide metabolism (Appendix A), and butanoate metabolism (Appendix A).

### 2.3. MetaMapp Network View of Significantly Altered Serum Metabolites in TCDD-Treated Mice When Compared with Vehicle Control Group

To further identify the mechanisms behind these metabolic shifts in the serum, metabolite enrichment based on structural homology was conducted using MetaMapp. MetaMapp analysis of the serum metabolites revealed that prostaglandins, sphingolipids, ethanolamines, and multiple vitamin analogs were selectively overabundant following TCDD exposure when compared with controls (Figure 3). 

### 2.4. Pathway Analysis of Pathologically Relevant Metabolites in the Serum 

IPA analysis was conducted to better understand the physiological importance of significantly altered metabolites within the serum and cecal content. In the serum, there were seven metabolites that showed increased presence following TCDD exposure when compared with the control group that either indirectly or directly showed associations with immunologically relevant proteins such as NF-κB, transforming growth factor-beta (TGF-β), and forkhead box protein 3 (FOXP3) (Figure 4a). Additionally, there were 8 direct and indirect interactions between metabolites that regulate the activity of ERK1/2, a master regulator of cell proliferation and cell death (Figure 4a). Heat map demonstrating metabolites with significantly modulated abundance in VEH- vs. TCDD-treated mice (Figure 4b). Heat map (Figure 4c) and bar charts (Figure 4d) demonstrating the abundance of all significantly modulated metabolites of interest identified using IPA following TCDD treatment are shown. 

### 2.5. Ontological Association of Dysregulated Serum Metabolites with Cellular Functionality and Disease Pathology

Next, we tried to correlate the changes in the metabolome composition following TCDD exposure in mice with diseases and disorders, molecular and cellular functions, and physiological system development and function, by using IPA software. The dysregulated metabolome profile within the serum of TCDD-exposed mice was most heavily correlated with the diseases and disorders to include cancer, organismal injury and abnormalities, developmental disorders, hereditary disorders, and metabolic diseases (Figure 5a). The molecular and cellular functions associated with the dysregulated serum metabolite profile included alterations in cellular growth and proliferation, cell cycle, protein synthesis, carbohydrate metabolism, and molecular transport (Figure 5b). The top enriched physiological system development and function processes based on the content within the serum included organismal development, endocrine system development and function, hematological system development and function, tissue development, and lymphoid structure and development (Figure 5c).

### 2.6. MetaMapp Network View of Significantly Altered Cecal Content Metabolites in TCDD-Treated Mice When Compared with the Vehicle Control Group

To further identify the mechanisms behind these metabolic shifts in the cecal contents, metabolite enrichment based on structural homology was conducted using MetaMapp. MetaMapp analysis of the cecal content metabolites revealed that nucleotides, lipids, ethanolamine, indoles, steroids, dipeptides, and phenylalanine intermediates were selectively modulated following TCDD exposure (Figure 6).

### 2.7. TCDD Exposure Alters Cecal Metabolome That Is Associated with Altered Immune Response

IPA analysis was conducted to better understand the physiological importance of significantly altered metabolites within the cecal content. In the cecum, there were two metabolites that showed increased presence following TCDD exposure when compared with the control group that either indirectly or directly showed associations with pro-inflammatory cytokines and NF-κB (Figure 7a). A heat map demonstrating metabolites with significantly modulated abundance in VEH- vs. TCDD-treated mice is depicted in Figure 4b. Heat map (Figure 4c) and bar charts (Figure 4d) demonstrating the abundance of all significantly modulated metabolites of interest identified using IPA and xanthosine, which was identified independently of IPA, are also shown. 

### 2.8. Ontological Association of Dysregulated Cecal Metabolites with Cellular Functionality and Disease Pathology

The disparity in metabolites within the cecal content is most heavily correlated with diseases and disorders including cancer, organismal injury and abnormalities, tumor morphology, endocrine system disorders, and metabolic disease (Figure 8a). The molecular and cellular functions associated with the dysregulated cecal content profile included altered abilities in small molecule biochemistry, cell death and survival, cellular assembly and organization, cellular function and maintenance, and cell morphology (Figure 8b). The top enriched physiological system development and function processes based on the contents within the cecum included nervous system development and function, endocrine system development and function, hematological system development and function, immune cell trafficking, and organismal development (Figure 8c).

## 3. Discussion

The composition of metabolites in both the serum and cecal content, where a large portion of host–microbe interactions occur, provides novel insights into the physiology behind TCDD-mediated toxicity. This study demonstrates in an animal model that acute TCDD exposure significantly modulates the metabolomic profile at these two sites. The differentially abundant metabolites present in the circulation and the GI tract reveal candidate pathways by which TCDD can potentially contribute toward toxicity including early development, as well as immunosuppression and cancer progression [29,30,31]. The significant change in the global composition of the serum metabolites and GI barrier site metabolites corroborates the previous conception that TCDD exposure drives metabolic imbalance within the host organism. Metabolic imbalance following TCDD exposure has been studied previously with a primary focus on membrane proteins involved in the degradation of adipose tissue, bile acid accumulation inducing hepatotoxicity, liver, and skeletal muscle metabolism, and long term effects of TCDD exposure on surviving individuals [14,32,33]. These previous studies indicated the potential role of the metabolome in the pathology associated with TCDD exposure and intuitively suggested studying the acute effects of TCDD exposure within an animal model to better understand the immediate progression of TCDD-mediated toxicity. 

Within the serum, we saw significant changes in various amino acid abundances, increases in various bioactive lipid concentrations, and an upsurge in arachidonic acid-derived prostaglandins that serve as biomarkers signifying organismal injury. Previous studies exploring circulating amino acid concentrations showed a significant increase associated with TCDD-mediated toxicity due to increased mobilization and reduced metabolism of amino acids in the liver [30]. This corresponded with significantly higher concentrations of L-phenylalanine, L-tyrosine, L-threonine, leucine, L-glutamate, N-acetyl glutamic acid, L-serine, methionine, L-lysine, and the proteogenic organic acid proline measured in this study. Additionally, previous findings suggested that TCDD exposure increases the biosynthesis of sphingolipids in normal human epidermal keratinocytes [34]. Circulating metabolites associated with sphingolipid biosynthesis, specifically the ceramide pathway, were all elevated in mice exposed to TCDD. These sphingolipids are protective molecules that mediate a wide variety of cellular functions that are associated with TCDD- induced toxicity including apoptosis, proliferation, inflammation, and necrosis [35,36]. Inflammation is also largely mediated by FAs and their products with omega-6 FAs associated with pro-inflammatory processes and omega-3 FAs associated with anti-inflammatory processes [37]. All four PUFAs were detected at higher concentrations in TCDD-exposed mice, suggesting that inflammatory responses may be upregulated. AA, an omega-6 PUFA, is upregulated following injury or irritation and metabolized into eicosanoids, an important group of inflammatory mediators [38,39,40]. DHA, and omega-3 PUFA, is a precursor for the distinct family of specialized pro-resolving mediators (SPMs) called protectins as well as other SPMs such as resolvins and maresins [41,42]. ALA, an omega-3 PUFA, has been associated with the reduction in pro-inflammatory gene expressions such as IL-6, cyclooxygenase 2 (COX2), and TNF-alpha [43]. Lastly, LA, an omega-6 PUFA, has been the topic of much controversy with several studies attributing inflammatory properties to it and others attributing no significant inflammatory properties. Additionally, increased uric acid in TCDD-exposed mice could induce inflammation via NF-κB, as observed in HepG2 cells [44]. 

The cecal metabolome exhibited substantially fewer differences as a whole between TCDD and VEH mice when compared with changes in the serum; however, the changes observed indicated that TCDD exposure may still yield immediate changes in intestinal inflammation that modulate gut microbiota composition and metabolism, as has been demonstrated with a variety of other AHR ligands [45]. Purine-mediated inflammatory responses have been extensively explored indicating an evolved extracellular role as danger signals released during events such as cell lysis, apoptosis, and degranulation [46]. Reductions in metabolites associated with lysine degradation, nicotinamide metabolism, and butanoate metabolism are all indicative of suppressed metabolism of bioactive molecules in the gut [47].

NF-κB is a transcription factor vital for perpetuating innate and adaptive immunity by inducing the expression of pro-inflammatory cytokines, enhancing immune cell fitness, and coercing differentiation of immune cells into activated pro-inflammatory subsets with potent effector functions [48]. Exposure to dioxins such as TCDD has previously been shown to inhibit NF-κB expression in an AHR-dependent manner, inducing sustained immunosuppression and immunotoxicity [49,50]. In the current study, we observed that the metabolomic profile following TCDD exposure may contribute to this immunotoxicity by increasing the relative abundance of NF-κB inhibitors, such as linolenic acid and oleoylethanolamide, in circulation [51,52,53]. It is noted, however, that TCDD exposure results in the increased abundance of two NF-κB activating metabolites in arachidonic acid and uric acid [44,54]. At the site of injury following cell death, arachidonic acid is cleaved from the phospholipid membrane and enzymatically oxygenated via COX or lipoxygenase pathways, producing inflammatory mediator molecules [38]. The observed increase in the abundance of prostaglandin molecules and arachidonic acid correlates with a previously established increase in COX-1 activity following low-dose TCDD exposure in vitro [55]. The increased presence of the prostaglandin F_2α_, also referred to as “dinoprost”, within the serum is a marker of injury and inflammation within the local host tissues [56]. This increase in arachidonic acid and dinoprost is likely due to organismal injury, apoptosis, and lesion formation following TCDD exposure, and is likely being perpetuated by the previously established increases in COX-1 activity that result from TCDD exposure [55,57]. Increases in uric acid content within the serum have already been shown in previous studies focusing on the metabolome of patients exposed to toxic AHR ligands with structural similarity to TCDD [58]. While uric acid also works as an activator of NF-κB in both mice and humans, it has been previously shown that reducing serum uric acid content with allopurinol treatment results in a significant decrease in the anti-inflammatory cytokine TGF-β1 [59,60]. These counteracting results make it difficult to distinguish if uric acid levels in the serum are contributing to inflammatory processes via NF-κB induction or alleviating inflammatory processes by maintaining and increasing TGF- β1 production. Collectively, however, these results suggest that the circulating metabolic profile within the serum content following TCDD exposure is likely contributing to the known immunomodulation by dioxins via inhibition of NF-κB activity with some potential lingering NF-κB activation resulting from COX-1 activity at sites of cell damage and hyperuricemia. 

Extracellular signal-related kinase 1/2 (ERK1/2) is a kinase enzyme that regulates a variety of cell activities including metabolism, proliferation, and survival [61]. Previous research has demonstrated that TCDD-mediated toxicity in the contexts of cancer and neuropathology is partially dependent on ERK1/2 signaling pathways [62,63,64]. Additionally, it has been shown that multiple inhibitors of ERK1/2 can reverse the observed effects of TCDD exposure by regulating the steady-state levels of the AHR and inhibiting cytochrome P450 family 1 subfamily member 1 activity [65]. These results collectively demonstrated that the observed changes in host physiology following TCDD exposure are at least partially dependent on downstream ERK1/2 signaling subsequently following AHR activation. The amino acid L-phenylalanine, which is increased within the serum of TCDD-treated mice, has been shown to increase the activation of ERK1/2 via allosteric interactions with calcium ion sensing receptors [66]. Circulating levels of linolenic acid are increased following TCDD exposure, coinciding with vascular inflammation specific to endothelial cells via ERK1/2 signaling [67]. This correlates with previous studies demonstrating that TCDD exposure worsens the incidence of atherosclerosis via the induction of vascular inflammation in a murine model [68]. Increased levels of circulating uric acid, as seen in the TCDD group, may result in nephropathy via ERK1/2 phosphorylation [69]. In one study, investigating hyperuricemia in rats, the pharmacological inhibition of ERK1/2 resulted in a significant reduction in the Smad3-dependent renal pathology [70]. Oleoylethanolamide is known to induce anti-inflammatory effects via ERK1/2 inhibition in a model of LPS-stimulated THP-1 cells in vitro [53], while within the hippocampi of rats, in a study focusing on alcohol abuse, oleoylethanolamide has been shown to selectively increase p-ERK1/2 levels relative to the vehicle alone [53,71]. These conflicting results due to the promiscuity of the ERK1/2 activation and inhibition following oleoylethanolamide supplementation make it difficult to deem in what manner and which site the metabolite is enacting upon ERK1/2 in vivo; however, its anti-inflammatory effects have been previously well established [72,73]. Based on our findings, which demonstrate that multiple metabolites significantly regulated following TCDD exposure can directly interact with the ERK1/2, in combination with previous literature demonstrating AHR and ERK1/2 codependency, we suggest that TCDD-induced changes in the metabolome contribute to systemic toxicity, renal pathology, inflammatory dysregulation, and atherosclerosis via activation of ERK1/2 [74]. 

It has previously been established that exposure to dioxins that act via AHR activation results in widespread changes in the local environment of the gut that, in turn, results in lasting effects on host physiology; however, the manner by which modulated gut microbiota composition can affect host immunity following TCDD exposure is not yet well described [12,75]. Mass spectrometry analysis reveals that within the cecal contents of TCDD exposed mice, there was an increase in the concentration of the anti-inflammatory molecule linoleoyl ethanolamide, which has previously been shown to limit production of pro-inflammatory cytokines and inhibit translocation of the pro-inflammatory transcription factor NF-κB [76]. TCDD exposure also decreased the abundance of ethylene glycol, a substance used to induce renal inflammation and activate the previously described transcription factor NF-κB [77,78,79]. The antiparallel increase in linoleoyl ethanolamide and decrease of ethylene glycol within the cecal contents 72 h after TCDD exposure suggests that the metabolic dysbiosis occurring within the GI tract likely contributes to the immunosuppressive phenotype of TCDD in a manner dependent on NF-κB inhibition. Intestinal concentrations of the inflammation biomarker xanthosine, a product of nucleobase deamination that arises from RNA damage at sites of inflammation, were shown to be significantly reduced within the cecal contents of TCDD exposed mice [80]. This decreased xanthosine abundance suggests that acute exposure to TCDD results in the suppression of pro-inflammatory processes within the GI tract under normal conditions. Collectively, these results in the gut demonstrate that acute exposure to TCDD results in shifts in the cecal metabolome, which suggests a localized inhibition of inflammatory processes. 

## 4. Materials and Methods

### 4.1. Mice

Six-week-old female wildtype (WT) C57BL/6 mice (Jackson Laboratories, Bar Harbor, ME, USA) were housed in an AAALAC-accredited specific-pathogen-free animal facility located at the grounds of the University of South Carolina School of Medicine for the entirety of all experiments. Mice within the facility were housed within polycarbonate cages containing cellulose fiber chips as bedding in a temperature and humidity-controlled environment. After a 2-week acclimatization period, the mice were divided randomly into two groups that would be administered a single 100 µL intraperitoneal injection containing sterile corn oil (VEH group) or intraperitoneal injection of 10 µg/kg TCDD suspended within sterile corn oil (TCDD group). At the 72 h time point following TCDD or VEH exposure, the mice were humanely euthanized by an overdose of inhaled isoflurane. 

### 4.2. Chemicals and Reagents

The following reagents were used during the course of the experiments and were purchased as follows: TCDD was kindly provided by Dr. Steve Safe (Institute of Biosciences and Technology, Texas A&M Health Science Center, College Station, TX, USA); isoflurane (Henry Schein, Melville, NY, USA); corn oil, methanol, chloroform, and ethanol (Fisher Scientific, Pittsburgh, PA, USA); LC–MS-grade methanol (MeOH), acetonitrile (ACN), water, and formic (Thermo Fisher Scientific, Waltham, MA, USA).

### 4.3. Cecal Content and Serum Processing

Cecal content and serum samples were immediately processed according to previously described methods for metabolomic profiling using liquid chromatography–mass spectrometry (LC–MS) with slight modifications [81]. Contents extracted from the cecum, ~25 mg was aliquoted into 1.5 mL tubes (Eppendorf, Hamburg, Germany) containing ~20 mg acid-washed glass beads (Sigma-Aldrich, St. Louis, MO, USA), extracted into ice-cold MeOH:water (1:1, *v/v*) on a QIAGEN TissueLyzer at 50 Hz for 10 min (QIAGEN, Hilden, Germany), and centrifuged at 12,000 rpm for 10 min. Blood was collected using a sterile 27 gauge needle from the hepatic portal vein. For blood sera, 20 µL aliquots were extracted by adding 180 µL cold MeOH, briefly vortexed, and incubated at −20 °C for 30 min for protein precipitation. The supernatants of both matrices were dried in a vacuum concentrator (CentriVap, Labconco, MO, USA) and resuspended in ACN:water (2:98, *v/v*) upon analysis.

### 4.4. Instrumental Analysis

Global metabolomics profiling of cecal content and serum extracts was performed on a high-resolution accurate-mass (HRAM) mass spectrometry-based platform. The system consisted of a Vanquish UHPLC and a Q Exactive mass spectrometer (Thermo Fisher Scientific, Waltham, MA, USA) interfaced with heated electrospray ionization (HESI) source coupling to a hybrid quadrupole-orbitrap mass analyzer operated at the mass resolution of 70,000 full width at half maximum (FWHM) with AGC target of 3e6 and max IT of 220 ms. Prior to mass detection, metabolites were chromatographically separated using a Waters Acquity UPLC HSS T3 column (reverse phase C18, 100 Å, 1.8 μm, 2.1 mm × 100 mm) (Milford, MA) kept at 40 °C. The mobile phases included 0.1% formic acid in water (A) and 0.1% formic acid in ACN (B), with a flow rate of 0.4 mL/min and gradient as follows: 2% B, 0–2 min; 2% to 80% B, 2–11 min; 80% to 98% B, 11–12 min; 98% B, 12–16 min; 98% to 2% B, 16–16.1 min; 2% B, 16.1–20 min. Mass spectrometric analysis was conducted under HESI positive mode under the following condition: sheath gas flow rate, 60 L/min; sweep gas flow rate, 1 L/min; aux gas flow rate, 10 L/min; aux gas heater temperature, 400 °C; capillary temperature, 325 °C; spray voltage, 2.75 kV. Stringent quality control/quality assurance (QA/QC) measures were taken throughout the analysis, spanning ion source cleanup, timely mass calibration, sample randomization, and intermittent QC injection (of pooled extracts). To identify structures of ion features of statistical significance, tandem mass spectra were further acquired under a hybrid mode of full scan MS1 alternating with parallel reaction monitoring (PRM) with the latter set at FWHM 17,500, AGC target of 2e5, and max IT 50 ms. 

### 4.5. Data Processing and Statistical Analysis

Full scan MS1 *.raw data were processed at XCMS Online (Scripps Research, La Jolla, CA, USA) for peak picking and peak alignment [82,83,84]. Principal component analysis (PCA) was conducted on resulting metabolites to produce scores plots visualizing separation between groups. Two-sided Welch’s *t*-test with a false detection rate (FDR) rate of 0.05 for α = 0.05, was carried out, and resulting metabolites were considered significant with q ≤ 0.05 and fold change (FC) ≥ 1.5. Ion features that reached significance were annotated using MS-FINDER 3.10 (Riken, Wako, Saitama, Japan). The annotated metabolites were then used to conduct pathway analysis on the MetaboAnalyst 3.0 platform (University of Alberta, Edmonton, AB, Canada) [85,86] using a hypergeometric test to determine metabolite enrichment and relative-betweenness centrality, to visualize pathway topology. Due to limitations in available identification databases for measured metabolites, only those with KEGG and/or HMDB identifiers were able to be used for this pathway analysis. MetaMapp analysis was conducted on significantly altered metabolites for the purpose of identifying structural similarities between modulated metabolites to better understand how acute TCDD exposure affects the composition of metabolite classes by chemical structure [87]. Ingenuity pathway analysis (IPA) was utilized for downstream effects on protein interactions and biochemical pathways based on significantly modulated metabolites [88]. The data shown in graphs display the mean ± standard error of the mean (SEM). GraphPad Prism 8 (GraphPad Software, San Diego, CA, USA) was used for statistical analysis. Degree of significance was demonstrated using the following key: * *p* < 0.05, ** *p* < 0.01, *** *p* < 0.001.

## 5. Conclusions

Previous studies on TCDD toxicity have primarily focused on the long-term effects of exposure on metabolite abundance or the immediate effects of exposure on host immunity. To define the unknown role of host metabolism in AHR-mediated toxicity following TCDD exposure, we applied mass spectrometry to measure the acute changes in serum and cecal metabolite abundance while demonstrating their contextual importance in pathologies that arise following TCDD exposure. Our work demonstrates that TCDD exposure results in significant modulations of the metabolomic profile of the serum and cecum contributing to previously established toxic side effects. Specifically, TCDD exposure triggers an increase in anti-inflammatory metabolites inhibiting NF-κB activity and ERK1/2 activators. The metabolome of TCDD-exposed mice suggests how TCDD contributes to the endured systemic immunosuppression and organismal injury while conserving the previously defined site-specific inflammation limited to vascular and renal tissue. Collectively, these results demonstrate that the changes in the metabolomic profile following TCDD exposure may play an important role in mediating TCDD toxicity by contributing to immunomodulation, tissue damage, and cell cycle regulation. Future research building upon this work that focuses on interventions correcting metabolic dysregulations following TCDD exposure may prove to be useful in limiting toxicity. 

## Figures and Tables

**Figure 1 ijms-22-11801-f001:**
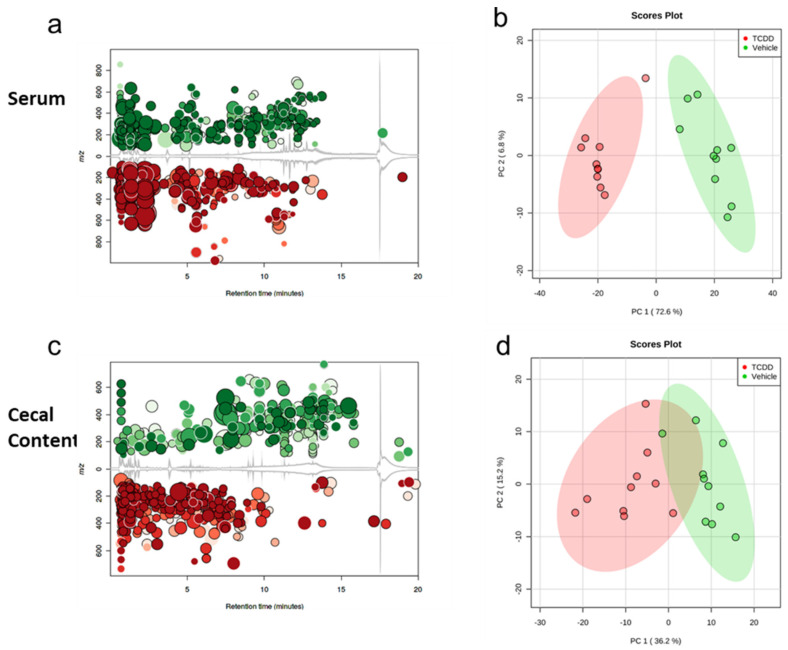
Changes in overall metabolic profile following acute TCDD exposure: (**a**) metabolomic global view of separation between TCDD vs. VEH. TIC cloud plot of serum metabolites distinguishing 1212 metabolites significantly altered (*p* ≤ 0.01). For these data, dot size correlates to the fold change in abundance between the TCDD (red)- vs. VEH (green)-treated mice, hue intensity refers to the *p* value significance, and the gray line demonstrates the mean intensity output per retention time and mass-to-charge ratio; (**b**) principal component analysis (PCA) plotting of serum metabolites, with 95% confidence region highlighted; (**c**) TIC cloud plot of cecal metabolites distinguishing 902 metabolites significantly regulated (*p* ≤ 0.01). For these data, dot size correlates to the fold change in abundance between the TCDD (red)- vs. VEH (green)-treated mice, hue intensity refers to the *p* value significance, and the gray line demonstrates the mean intensity output per retention time and mass-to-charge ratio; (**d**) principal component analysis (PCA) plotting of cecal metabolites, with 95% confidence region highlighted.

**Figure 2 ijms-22-11801-f002:**
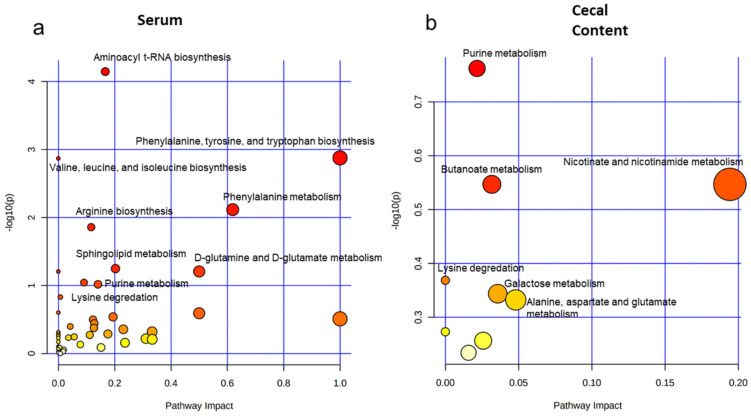
Identification of metabolic pathways most impacted following acute TCDD exposure. Pathway analysis based on overrepresentation analysis (hypergeometric test) and pathway topology analysis (relative-betweenness centrality) reveals dysregulated upstream pathways of observed metabolites in acute TCDD exposure. MetaboAnalyst analysis revealed 40 unique metabolic pathways altered following TCDD exposure in the serum (**a**) and nine unique metabolic pathways altered following TCDD exposure in the cecum (**b**). Analysis was performed in MetaboAnalyst, using “Mammals: Mus musculus (KEGG)” for pathway reference. (Larger dot size correlates to the relative pathway impact and intensity of the red hue correlates to the *p* value significance).

**Figure 3 ijms-22-11801-f003:**
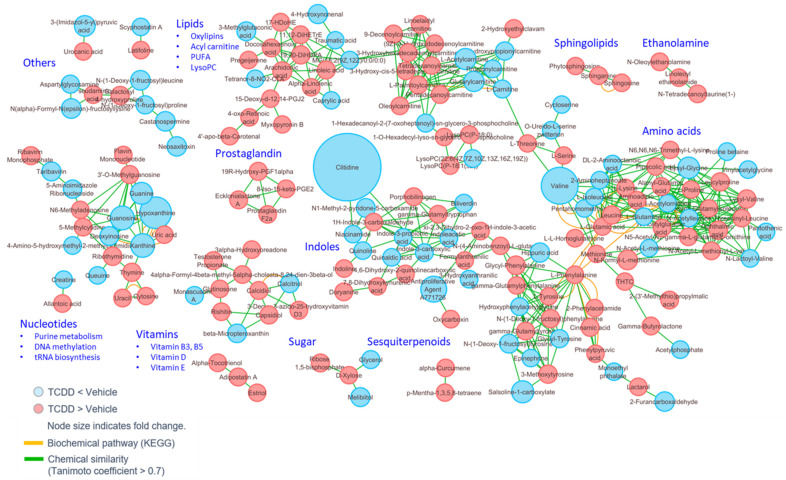
MetaMapp network view of significantly changed serum metabolites (*p* < 0.05) in TCDD-treated mice, as compared with the vehicle control group. The clustering is constructed based on biochemical relationships (KEGG) and chemical similarity (Tanimoto coefficient > 0.7).

**Figure 4 ijms-22-11801-f004:**
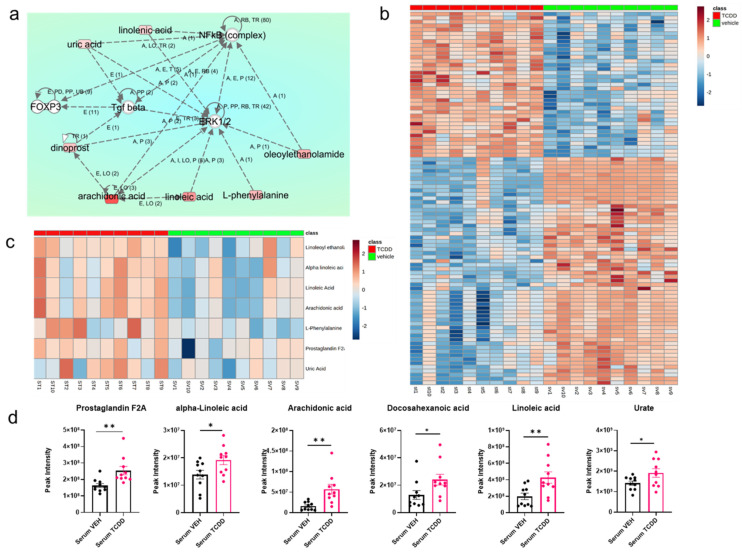
Pathway analysis of pathologically relevant metabolites in the serum. TCDD exposure significantly regulates the circulating metabolome in an immunosuppressive manner: (**a**) ingenuity pathway analysis of significantly modulated serum metabolites (metabolites in green are higher in the VEH-treated mice, while those in red are higher in the TCDD-exposed mice); (**b**) heat map displaying the intensity of significantly differentially expressed individual metabolites within the serum of VEH and TCDD mice (*n* = 10 per group). Heat maps display the normalized intensity by z-score for each metabolite; (**c**) heat map displaying significantly modulates metabolites of interest within the serum of VEH and TCDD mice. Heat maps display the normalized intensity by z-score for each metabolite; (**d**) peak intensities identified via LC-HRMS profiling of metabolites of interest that are differentially abundant between the serum of VEH and TCDD mice (Prostaglandin F2A t(18) = 2.873, ** *p* = 0.01; Alpha-linolenic acid t(18) = 2.877, * *p* = 0.03; Arachidonic Acid t(18) = 3.390, ** *p* = 0.003; Docosohexanoic Acid t(18) = 2.206, * *p* = 0.04; Linoleic Acid t(18) = 2.877, ** *p* = 0.01; Urate t(18) = 2.127, * *p* = 0.047).

**Figure 5 ijms-22-11801-f005:**
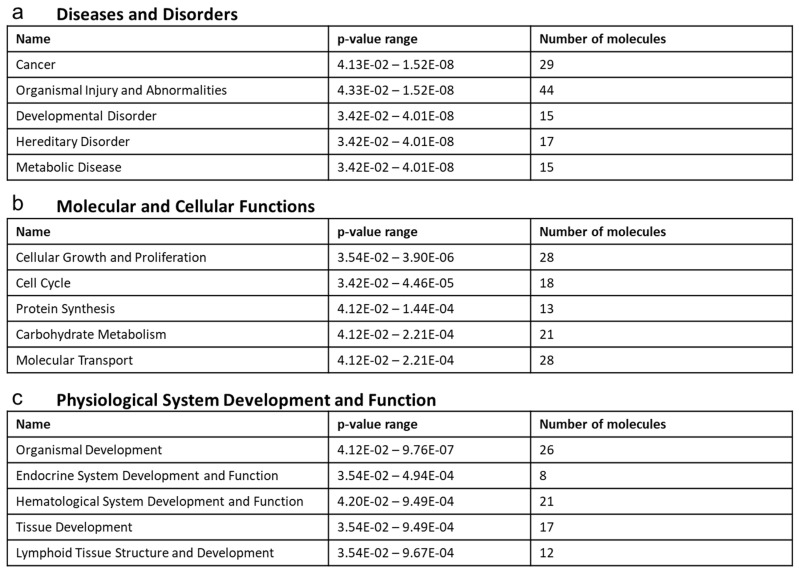
Ontological association of dysregulated serum metabolites with cellular functionality and disease pathology. Top enriched diseases and biological functions associated with a dysregulated metabolomic profile within the serum following TCDD treatment using ingenuity pathway analysis: (**a**) the top five diseases and disorders, (**b**) molecular and cellular functions, and (**c**) physiological system development and function associations are listed.

**Figure 6 ijms-22-11801-f006:**
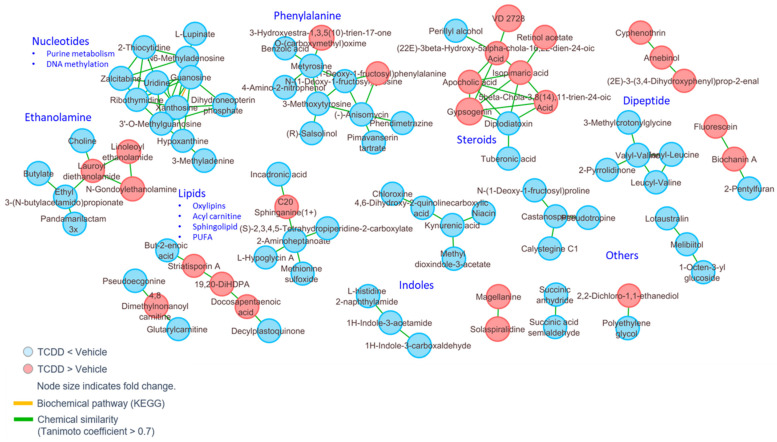
MetaMapp network view of significantly changed cecal content metabolites (*p* < 0.05) in TCDD-treated mice, as compared with the vehicle control group. The clustering is constructed based on biochemical relationships (KEGG) and chemical similarity (Tanimoto coefficient > 0.7).

**Figure 7 ijms-22-11801-f007:**
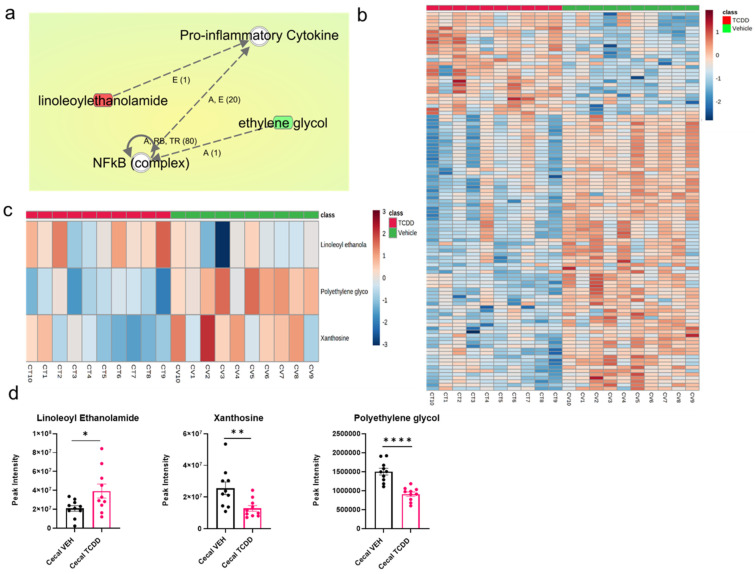
Pathway analysis of pathologically relevant metabolites in the cecal contents. TCDD exposure promotes immunosuppression via modulations in the cecal metabolome: (**a**) ingenuity pathway analysis of significantly modulated cecal metabolites (metabolites in green are higher in the VEH-treated mice, while those in red are higher in the TCDD-exposed mice); (**b**) heat map displaying an abundance of individual metabolites within the cecal contents of VEH and TCDD mice. Heat maps display the normalized intensity by z-score for each metabolite; (**c**) heat map displaying significantly modulated metabolites of interest within the cecal contents of VEH and TCDD mice. Heat maps display the normalized intensity by z-score for each metabolite; (**d**) peak intensities identified via LC-HRMS profiling of metabolites of interest that are differentially abundant between the cecal contents of VEH and TCDD mice (Linoleoyl Ethanolamide t(18) = 2.307, * *p* = 0.033; Xanthosine t(18) = 2.895, ** *p* = 0.009; Polyetheylene Glycol t(18) = 5.705, **** *p* < 0.0001).

**Figure 8 ijms-22-11801-f008:**
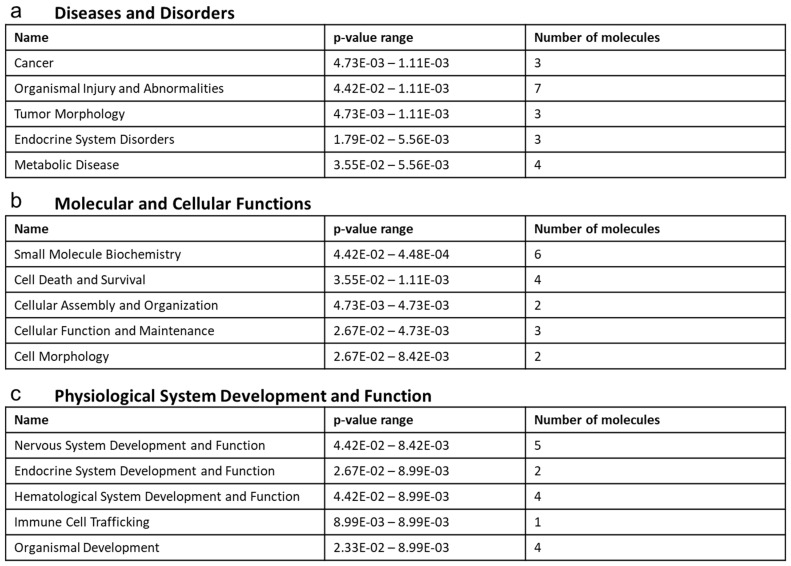
Ontological association of dysregulated cecal metabolites with cellular functionality and disease pathology. TCDD exposure promotes immunosuppression via modulations in the cecal metabolome. Top enriched diseases and biological functions associated with a dysregulated metabolomic profile within the cecal content following TCDD treatment using ingenuity pathway analysis. The top five (**a**) diseases and disorders, (**b**) molecular and cellular functions, and (**c**) physiological system development and function associations are listed.

## Data Availability

The datasets presented in this study can be found in online repositories hosted via https://www.ncbi.nlm.nih.gov/ (accessed on 30 August 2020) under the bioproject number PR001181. The bioproject data are also available at Metabolomics Workbench: https://www.metabolomicsworkbench.org/data/DRCCMetadata.php?Mode=Project&ProjectID=PR001181 (accessed on 30 August 2020).

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
