# Peer review of "Effects of Acute 2,3,7,8-Tetrachlorodibenzo-p-Dioxin Exposure on the Circulating and Cecal Metabolome Profile"

_ijms, 2021, doi:10.3390/ijms222111801_

Round 1

Reviewer 1 Report

Summary

The submitted manuscript examined the acute metabolomic changes in the serum and cecum elicited by 2,3,7,8-tetrachlorodibenzo-p-dioxin (TCDD). The authors used mass spectrometry to evaluate metabolite levels in 10 ug/kg TCDD treated female mice after 72 hours and demonstrate numerous changes in the metabolomic profiles. The studies show that the cecum had fewer changes in metabolite levels and that metabolites which modulate NF-KB signaling are altered such as uric acid and arachidonic acid. Collectively, the finding of the manuscript would be of interest to readers of IJMS but the authors should make significant revisions to their figures and legends to more clearly present their data.

Comments.

  • The abbreviation VEH for vehicle is used in the abstract but not defined.
  • The authors indicate in a couple of instances that no acute metabolomic studies have been performed in TCDD treated mice. I am aware of at least one that the authors may want to be made aware of (PMC3243745).
  • Figure 1 legend: template text not removed “Schemes follow another format. If there are multiple panels, they should be listed as:”. The gray line, point shading, and point size of individual points is not defined.
  • Figure 3. The size of the nodes is not defined. It may be due to resolution, but it is extremely difficult to distinguish between the yellow and green lines.
  • Figure 4. Is panel C not redundant with panel D? What are the units for the heat maps? Why is the metabolite identity, class, or other piece of information not shown in panel B? As it is, is panel B not just redundant with Figure 1 which shows that some metabolites are induced and others are repressed? The authors should consider more closely their choice of visualizations to best present their data in a manner that is of value to readers.
  • The same comments from the serum figures also apply to the cecum figures.
  • Line 323: Citation format is inconsistent for Wu et al. 2011.
  • Materials and methods: Please specific how the samples were collected. Where was the blood collected from and was there any additive in the collection tubes such as EGTA or sodium citrate? Were the cecal contents removed from the cecum prior to metabolomics analysis? Were sample frozen or immediately processed?
  • Conclusions: The authors state “site-specific inflammation limited to vascular and renal tissue”. What do the authors mean as there is a lot of evidence that TCDD also elicits inflammation in the liver.
  • I commend the authors for having submitted their data to metabolomics workbench which is critical for research reproducibility. Can the authors clarify in the manuscript that the data is available on metabolomics workbench specifically rather than just an NIH repository?
  • It is unclear if the processed metabolomic data is provided as supplementary material. It would be of benefit to readers. The supplementary material section simply states “title” (lines 455-456). This section actually appears to be a place holder as not all supplementary figures are listed and a video is listed but not mentioned anywhere in the manuscript.
  • General comment. While the evidence for metabolites affecting the NF-KB pathway and ERK1/2 is supported by the data. It is unclear why the authors chose to focus on that pathway. Was it among the most affected pathway, or was the evaluation of this pathway hypothesis driven? Why not discuss the most affected pathways if it wasn’t the inflammatory pathways?

Author Response

First off, we would like to thank the reviewers for taking the time to read our article and provide invaluable feedback to improve the quality of our submission.

Reviewer 1:

Comment: The abbreviation VEH for vehicle is used in the abstract but not defined.

Response: This definition for this abbreviation has been added to line 21

Comment: The authors indicate in a couple of instances that no acute metabolomic studies have been performed in TCDD treated mice. I am aware of at least one that the authors may want to be made aware of (PMC3243745).

Response: We thank the reviewer for pointing this out.  This reference has been cited and we have changed our statement to:  attributed to a limited number of studies dedicated to studying the direct changes

Comment: Figure 1 legend: template text not removed “Schemes follow another format. If there are multiple panels, they should be listed as:”. The gray line, point shading, and point size of individual points is not defined.

Response: Thanks for catching the template text, this has been removed (line 88). This issue appears to have occurred here as well as with the mention of supplemental figures which were uploaded separately via ppt. Figure legend 1 has been updated to define what these data mean (line 88-98)

Comment: Figure 3. The size of the nodes is not defined. It may be due to resolution, but it is extremely difficult to distinguish between the yellow and green lines.

Response: We have reuploaded this figure with increased definition. Please let us know if this is now acceptable

Comment: Figure 4. Is panel C not redundant with panel D? What are the units for the heat maps? Why is the metabolite identity, class, or other piece of information not shown in panel B? As it is, is panel B not just redundant with Figure 1 which shows that some metabolites are induced and others are repressed? The authors should consider more closely their choice of visualizations to best present their data in a manner that is of value to readers.

Response: Panel C demonstrates normalized intensity by virtue of z-score, while panel D displays the raw peak intensity collected from MS reads. The figure legend has been modified to signify this. While they both display change in metabolite abundance, panel D is not normalized to the mean and instead demonstrates the raw metabolite abundance. We feel that demonstrating both metrics for metabolite abundance is important. 

Comment: The same comments from the serum figures also apply to the cecum figures.

Response: Figure 7 legend has also been modified, as has the definition of Figure 6.

Comment: Line 323: Citation format is inconsistent for Wu et al. 2011.

Response: This citation has been modified to be consistent with the article

Materials and methods: Please specific how the samples were collected. Where was the blood collected from and was there any additive in the collection tubes such as EGTA or sodium citrate? Were the cecal contents removed from the cecum prior to metabolomics analysis? Were sample frozen or immediately processed?

Response: For serum collection no anti-coagulation additives were used, such as EDTA or Na Citrate. Contents were extracted from the cecum, this has been updated in line 395-406. Blood was collected via the hepatic portal vein.

Conclusions: The authors state “site-specific inflammation limited to vascular and renal tissue”. What do the authors mean as there is a lot of evidence that TCDD also elicits inflammation in the liver.

I commend the authors for having submitted their data to metabolomics workbench which is critical for research reproducibility. Can the authors clarify in the manuscript that the data is available on metabolomics workbench specifically rather than just an NIH repository?

Response: The data availability statement has been updated to include the metabolomics workbench link to the project

Comment: It is unclear if the processed metabolomic data is provided as supplementary material. It would be of benefit to readers. The supplementary material section simply states “title” (lines 455-456). This section actually appears to be a place holder as not all supplementary figures are listed and a video is listed but not mentioned anywhere in the manuscript.

Response: The supplemental figures were uploaded as a separate ppt. file for IJMS submission processes, we have modified this section to include the SF titles (lines 471-478). Thank you for noticing this. A link for the SFs has not been supplied to us, please contact us or the editor for the supplement if you are not able to view it.

General comment. While the evidence for metabolites affecting the NF-KB pathway and ERK1/2 is supported by the data. It is unclear why the authors chose to focus on that pathway. Was it among the most affected pathway, or was the evaluation of this pathway hypothesis driven? Why not discuss the most affected pathways if it wasn’t the inflammatory pathways?

Response: Of the pathways with substantial interaction in IPA we identified the ones with immunological implications, which were amongst the top regulated pathways. Additionally, since TCDD provides potent acute immunosuppression at this dose, we aimed to define how the metabolome contributes to this effect. We believe that the release of these data in a repository will allow for the potential of meta-analysis by other groups to properly identify and define any important immunity independent effects of acute TCDD exposure.

Reviewer 2 Report

This manuscript by Dopkins et al. entitled “Effects of acute 2,3,7,8-Tetrachlorodibenzo-p-dioxin (TCDD) exposure on the circulating and cecal metabolome profile” describes the comparison of metabolomic profiles between circulating blood and cecum in mice underwent acute exposure to TCDD. The manuscript is well written, and I found no particular typos and other unclear sentences. The authors provide remarkable metabolomic changes in metabolism of bioactive lipids, amino acids, and carbohydrates in blood resulting from the modulation of NF-kB, and ERK1/2 signaling. Furthermore, the authors focus on cecal metabolome to provide the metabolomic changes due to changes in metabolism of the gastrointestinal microbiota. One important conclusion is that these changes are derived from regulation of host immunity and inflammation. I would like to raise general comments.

Major question

  1. I admit there are few studies done for pathological and biochemical analysis after acute TCDD exposure. TCDD will, however, never be used for any purpose of evil intention. I would like the authors to explain why research on “high acute” TCDD-dosed mice is meaningful. I am wondering if the observed profiles in metabolome changes after acute TCDD-dosing might be similar to other class of xenobiotics. If so, the current study bears large scientific significance.

  1. The authors state that metabolomic changes in cecum are due to changes in metabolism of gastrointestinal microbiota. How did authors conclude it? I am wondering if it may be due to changes in metabolism of epithelial cells of gastrointestinal tracts.

  1. If the metabolomic changes in cecum is attributed to microbiota, then, the blood metabolomic changes might be resulted from gastrointestinal microbiota. Is my guess correct?

  1. Are ethical considerations on animal experiments described anywhere in this study?

Minor question

  1. In Discussion section, at lines 289-291 on Page 10 of 18, “The observed increase in the abundance of prostaglandin molecules and arachidonic acid correlates with a previously established increase in COX-1 activity and cell death following TCDD exposure [54].” This is a reference, but I would like to authors to state whether this observation is after high acute TCDD dosing or not.

Author Response

First off, we would like to thank the reviewers for taking the time to read our article and provide invaluable feedback to improve the quality of our submission.

Major questions:

Comment: I admit there are few studies done for pathological and biochemical analysis after acute TCDD exposure. TCDD will, however, never be used for any purpose of evil intention. I would like the authors to explain why research on “high acute” TCDD-dosed mice is meaningful. I am wondering if the observed profiles in metabolome changes after acute TCDD-dosing might be similar to other class of xenobiotics. If so, the current study bears large scientific significance.

Response: We wish to point out that TCDD was indeed used to poison Ukrainian President Victor Yushchenko in 2004 and it was concluded that routine techniques to detect TCDD metabolites should be developed to diagnose and treat such poisoning.  Also, during Vietnam War, there was significant exposure to TCDD through agent orange contamination.  Overall, our studies suggest the potential impacts of acute exposure to a potent AhR agonist.  While we believe that such changes in metabolome are likely to be seen with other xenobiotic AhR ligands such as  3-Methylcholanthrene, it would be interesting to study the effect of other AhR ligands such as dietary and endogenous, some having high affinity like TCDD. 

Comment: The authors state that metabolomic changes in cecum are due to changes in metabolism of gastrointestinal microbiota. How did authors conclude it? I am wondering if it may be due to changes in metabolism of epithelial cells of gastrointestinal tracts.

Response: This is a very good point. AHR ligands have been shown to alter GI microbiota contents and GI epithelium activity in manners that then interact with one another, however we did not demonstrate that these effects are due solely to either the microbiota or epithelium. We believe that the direct effects are initially enacted on the epithelium / local immune system, which then yields a differential cecal metabolome stemming primarily from the microbiota. We have altered the wording at points in the document to better reflect that these changes observed in the GI tract as they are not defined to definitively be microbiota dependent.

Comment: If the metabolomic changes in cecum is attributed to microbiota, then, the blood metabolomic changes might be resulted from gastrointestinal microbiota. Is my guess correct?

Response: We believe the changes in the cecum do impact the circulating metabolome, yes.

Comment: Are ethical considerations on animal experiments described anywhere in this study?

Response: The protocols were approved by Institutional Animal Care and Use Committee (IACUC).  

Minor questions:

Comment: In Discussion section, at lines 289-291 on Page 10 of 18, “The observed increase in the abundance of prostaglandin molecules and arachidonic acid correlates with a previously established increase in COX-1 activity and cell death following TCDD exposure [54].” This is a reference, but I would like to authors to state whether this observation is after high acute TCDD dosing or not.

Response: Citation 54 has been replaced with the proper citation. This miscue appears to have arisen at some point in the editing process. The proper citation demonstrating the increase in COX-1 activity following TCDD exposure has been reinstated. It was demonstrated using low dosage in vitro, this has been added to line 301. Thank you for noticing this crucial issue.